# Dietary Supplements on Controlling Multiple Sclerosis Symptoms and Relapses: Current Clinical Evidence and Future Perspectives

**DOI:** 10.3390/medicines6030095

**Published:** 2019-09-12

**Authors:** Christina Tryfonos, Maria Mantzorou, Dimitris Fotiou, Michael Vrizas, Konstantinos Vadikolias, Eleni Pavlidou, Constantinos Giaginis

**Affiliations:** 1Department of Food Science and Nutrition, University of the Aegean, Myrina, 81400 Lemnos, Greece; mantzorou.m@aegean.gr (M.M.); elenpav@aegean.gr (E.P.); cgiaginis@aegean.gr (C.G.); 2Department of Neurology, School of Medicine, Aristotelian University of Thessaloniki, 54624 Thessaloniki, Greece; dfotiou@med.auth.gr; 3Department of Neurology, School of Medicine, Democritus University of Thrace, 68100 Alexandroupoli, Greece; vrizasmichael@yahoo.com (M.V.); kvadikol@med.duth.gr (K.V.)

**Keywords:** dietary supplements, multiple sclerosis, inflammation, antioxidant, clinical studies

## Abstract

**Background:** Multiple sclerosis (MS) constitutes a chronic progressive demyelinating disease which negatively affects the central nervous system. MS symptoms detrimentally affect the quality of life, as well as the life expectancy of MS patients. In this aspect, the present study aims to critically summarize and evaluate the currently available clinical studies focusing on the potential beneficial effects of dietary supplements on controlling MS symptomatology and relapse. **Methods:** PubMed database was comprehensively searched, using relative keywords to identify clinical trials that investigated the beneficial effects of dietary supplementation against MS symptomatology and progression. 40 clinical trials were found, which were divided into categories. **Results:** Nutritional status of MS patients, as well as supplementation have been suggested as potential factors affecting progression. Several substantial studies have documented a systematically high prevalence of vitamin A, B12 and D3 deficiency amongst MS patients. At present, clinical data have suggested that most of the dietary supplements under study may exert antioxidant and anti-inflammatory properties, improving depression symptomatology and quality of life overall. However, malnutrition risk in MS patients has not been adequately explored in order for more precise conclusions to be drawn. The supplements that may have a positive effect on MS are vitamins, fatty acids, antioxidants, phytochemicals and melatonin. **Conclusions:** Several dietary supplements may decrease inflammation and fatigue, also increasing also autoimmunity tolerance in MS patients, and thus improving quality of life and life expectancy. Currently, there is no effective clinical indication for applying dietary supplementation as complementary treatment against MS symptomatology.

## 1. Introduction

Multiple sclerosis (MS) constitutes a chronic, relapsing, inflammatory condition of the central nervous system (CNS). It is characterized by destruction of myelin and subsequent deposits of scar tissue, and results in debilitating physical and cognitive deficits, as well as a substantial burden on quality of life (QOL). The most common symptoms comprise paresthesia, numbness or weakness of the limbs, diplopia, vision loss, ataxia, fatigue, bowel or bladder dysfunction, spasticity, and mental changes [1]. Although treatment advances have led to improved longevity, overall MS-attributed mortality rates have not changed over time [2]. Further, a recent report has supported evidence that the prevalence of MS is gradually increasing, particularly in women [2]. It is the most common demyelinating condition of the CNS affecting an estimated 2.5 million people worldwide and an estimated 100 thousand in Canada [3].

MS diagnosis is commonly based on the McDonald criteria, which aim to determine the presence of demyelinating lesions. Further criteria are consisted of the combination of clinical assessment, neurological examination (including nerve conduction studies), medical imaging, and spinal fluid analysis. Disease presentation varies substantially depending on the MS phenotype (i.e., relapsing-remitting (RRMS), primary progressive, secondary progressive, progressive-relapsing, clinically isolated syndrome) in terms of symptoms, pace, and progression [4]. The most common phenotype is RRMS, which is associated with alternating bouts of relapse and remission. Progressive MS is characterized by consistently worsening symptoms and some times and the disability. Disability is often quantified according to the Kurtzke Expanded Disability Status Scale (EDSS), which is a standardized measure that considers neurological and functional aspects of the disease [5]. The main causes of MS remain unknown Minor familial tendencies, geographic susceptibility, and viral infections (e.g., Epstein-Barr) have been highlighted as potential triggers. MS is more common in young adults, women, smokers, obese individuals, and individuals who live farther from the equator [3]. Latitude is inherently tied to sun exposure, and consequently, vitamin D status has also been considered to be a potential risk factor for MS, as well as a potential therapeutic option [6].

The common conventional therapy for MS contains certain immune modulating drugs to reduce the relapse frequency, glucocorticoids for the treatment of acute exacerbations, and amantadine for fatigue treatment. There are also several other medications for more specific MS-related symptoms [7]. Treatment effectiveness is difficult to ascertain in MS due to fluctuations in symptoms and frequent relapse and remission periods. Moreover, most treatments aim to maximize recovery from relapses, prevent fatigue and infection, and postpone bedridden stages of disease, as no proven treatments exist for changing the course of MS. Physical therapy may address functional disabilities, and pharmaceuticals may address spasticity and immunological symptoms [1].

The last years, becoming more interest the relationship between the dietary supplementation and multiple sclerosis (MS). This has focused largely on etiology and the influence of dietary supplements as a crucial factor in the amelioration of MS symptomatology and improvement of psychological status and life expectancy. While the cause of MS remains unknown and the influence of dietary supplements is unclear, recent studies on antioxidant and dietary supplements intake in MS are strengthening the rationale in support of a healthy eating regime following diagnosis (Figure 1).

The aim of this review was to collect and critically summarize the current literature on the effect of supplementation on MS symptomatology and progression. Forty clinical trials were found in PubMed database, after using relative keywords to identify clinical trials that investigated the beneficial effects of dietary supplementation against MS symptomatology and progression. The clinical trials were divided into categories based on the type of supplements (Table 1).

## 2. Dietary Supplements and Multiple Sclerosis

### 2.1. Fatty Acids

#### 2.1.1. Omega-6 Fatty Acids

Omega-6 essential fatty acids (EFAs) exert a crucial role in both the synthesis and metabolism of myelin. MS patients showed significantly lower mean concentration of serum linoleic acid, lymphocytes, and cerebrospinal fluid (CSF) compared to healthy individuals in most [23,24,25], but not all [26] currently available studies. CSF can show the changes on the CNS CSF analysis provides insight on the state of health of the CNS, and it can help investigate the mechanisms of CNS injury and repair [27]. The decreased linoleic acid concentrations may be ascribed to malabsorption, since 42 out of 52 MS patients showed increased fecal fat excretion [28]. However, in another study, MS patients showed normal EFAs absorption. It should be noted that low linoleic acid concentrations are not specific to MS. However, certain MS patients may be at high risk of omega-6 fatty acid (ω-6 FA) deficiency [29]. This possibility has been supported by the fact that erythrocytes and lymphocytes of MS patients are characterized by abnormal electrophoretic mobility [29]. This irregularity was inverted by primrose oil supplementation, which contains ω-6 FA such as linoleic acid and gamma-linolenic acid [29].

Several clinical studies have also evaluated the effect of supplementation with linoleic acid in MS patients. Sunflower oil derived Linoleic acid or sunflower oil from other sources and was administered at doses of 11.5 g/day to 23 g/day for 2 to 2.5 years, while control groups received olive oil (OO) or oleic acid. In most studies, the treatment group showed fewer or less severe relapses compared to the control group [8]. However, one study did not find any benefit [30]. A pooled analysis of several clinical trials showed that linoleic acid considerably reduced the deterioration rate in patients with minimal or no disability at the beginning of the study, and both the severity and duration of relapses of patients regardless of baseline disability and illness duration [31].

Further research on these supplements and the MS should be continued with new and bigger clinical trials.

#### 2.1.2. Omega-3 Fatty Acids

Omega-3 fatty acids (ω-3 FA) exert crucial role in normal brain function and the CNS [9]. Moreover, ω-3 FA deficiency has been considered to increase the susceptibility to damage of myelin. Notably MS patients showed significantly lower levels of eicosapentaenoic acid (EPA) in erythrocyte phospholipids, as well as lower docosahexaenoic acid (DHA) concentration in adipose tissue compared to healthy individuals [9].

In a double-blinded clinical study, fish oil (FO) supplementation led to clinical benefit of borderline statistical significance. In fact, 112 RRMS patients were randomized to receive 10 g/day of FO or placebo (OO) for a period of 2 years. Patients were advised to reduce animal fat intake and to increase dietary ω-6 polyunsaturated fatty acids intake. After 2 years, 51% of patients in the treatment group and 41.4% of patients in the placebo group showed better or the same scores, according to the Kurtzke Disability Status Scale [9].

In a similar study, Weinstock-Guttmana B. et al. investigated whether a low fat diet with ω-3 FA may improve QOL in RRMS patients [10]. In this 1-year, double-blinded clinical study, RRMS patients were randomized to 2 dietary intervention groups: One group received a low fat diet (with 15% fat) with ω-3 FO supplements, and the second group received the AHA Step I diet (with fat 30%), while receiving OO supplements. In this study, 31 RRMS patients were enrolled and clinical benefit for the FO group at 6 months was observed. Also, decreased fatigue on the OO group at 6 months was noted. Both groups had reduced relapse rates compared to the rates during the 1 year prior to the study. This study supported evidence that a low fat diet enhanced with ω-3 PUFA may exert moderate benefits in RRMS patients adjunct to pharmacological therapy.

Another study, concerning ω-3 FA designed from Shinto et al. evaluated the effect of ω-3 FA on matrix metalloproteinase-9 (MMP-9) production by immune cells in MS patients [11]. It has been already known than MMPs play an important role in the immunopathogenesis of MS, partly via the disruption of the blood-brain barrier and the recruitment of inflammatory cells into the CNS. Moreover, MMPs can also enhance the cleavage of myelin basic protein and the demyelination process [32]. In an open-label clinical study, fatty acid levels and QOL were assessed in 10 RRMS patients who received ω-3 FA supplementation at 9.6 grams/day FO. RRMS patients were evaluated at baseline, after 1 and 3 months of ω-3 FA supplementation, and after a 3-month wash out period. ω-3 FA supplementation significantly decreased MMP-9 levels in RRMS patients, while their immune cell secretion of MMP-9 was considerably reduced by 58% after 3-months of ω-3 FA supplementation compared to baseline. At the same time, a statistically significant increase in ω-3 FA levels in red blood cell membranes was recorded [11].

Although benefits have been observed after ω-3 FA supplementation in MS patients, a further investigation of the association between MS symptoms and ω-3 FA should be undertaken.

### 2.2. Coenzyme Q10

A recent study by Sanoobar et al. investigated the effect of coenzyme Q10 (CoQ10) supplementation on depression and fatigue in MS patients. 48 RRMS patients were enrolled in a 12-week placebo-controlled randomized double-blinded clinical study where they received 500 mg CoQ10 daily, or placebo for 12 weeks. Fatigue symptoms were evaluated according to the fatigue severity scale (FSS), while the Beck depression inventory (BDI) questionnaire was applied to assess depressive symptoms [12]. A significant decrease on FSS was found in the CoQ10 group during the intervention period. Moreover, while a significant increase on FSS within the placebo group changes were also recorded. A significant time-by-treatment interaction for FSS and BDI was observed, revealing significant decrease of FSS, signifying an improvement concerning fatigue symptoms, and of BDI in the CoQ10 group compared to the placebo group thus supporting evidence that CoQ10 supplementation at a dose of 500 mg/day may improve fatigue and depression in MS patients [12]. In an earlier published study, Sanoobar et al. investigated the effect of coenzyme Q10 supplementation on inflammatory and anti-inflammatory markers in the same RRMS patient group, showing that CoQ10 supplementation may also reduce the inflammatory markers TNF-α, IL-6, and MMP-9. Half of the patients were randomised to the placebo group and half in the 500 mg/day CoQ10 supplementation group for 12 weeks. Peripheral blood samples were collected at baseline and after 12 weeks, to measure inflammatory (tumor necrosis factor-α (TNF-α), interleukin (IL)-6, and MMP-9 and anti-inflammatory (IL-4) and Transforming growth factor beta (TGF-β) markers. After 12 weeks of supplementation, a significant decrease of TNF-α levels in the CoQ10 group was found. RRMS patients in the CoQ10 group presented significantly lower IL-6 levels compared to the placebo group. CoQ10 supplementation also resulted in decreased serum MMP-9 levels compared to the placebo group. However, CoQ10 supplementation did not affect either IL-4 or TGF-β levels [13]. Considering the effect of CoQ10 supplementation on oxidative stress and antioxidant enzyme activity in the same RRMS patient group, malondialdehyde (MDA), total antioxidant capacity (TAC) and antioxidant enzymes (superoxide, dismutase (SOD), glutathione peroxidase (GPx)) activity were evaluated in fasting blood samples before and after a 12-week intervention. Notably, CoQ10 supplementation at a dose of 500 mg/day resulted in a decrease of oxidative stress and increase antioxidant enzyme activity in RRMS patients [14].

The above results are seem interesting, but they need more clinical trials on a larger population on MS patients.

### 2.3. Vitamin B7 (Biotin)

Biotin is a vitamin acting as a coenzyme for carboxylases, being involved in several crucial steps of energy metabolism and fatty acids synthesis. Among others, biotin activates acetyl-coA carboxylase, which is a key rate-limiting enzyme during the synthesis of myelin [15]. It is known that in MS there is a decreased synthesis of myelin. High biotin doses in MS patients could exert a positive effect for MS patients [33]. Sedel F. et al. assessed the clinical efficacy and safety of high biotin doses in progressive MS patients. In this pilot study, high biotin doses (100–300 mg/day) were administered to 23 consecutive patients with primary and secondary progressive MS for a period from 2 to 36 months. This study supported evidence that high biotin doses may exert a positive effect on disability and progression in this MS patient population [15].

In the small study by Tourbah et al. [16] some progressive MS patients treated with high dose biotin showed improvement. However another recent study did not show a positive effect [34]. On the contrary, although high doses were safe, no long-term benefit was observed, while deterioration was noted in one third of the progressive MS patients during supplementation. It is also important to mention the case study by Maillart et al. who observed severe transient myopathy in a patient with progressive MS on high-dose biotin [35].

A lot of benefits have been observed after biotin supplementation in MS patients but the clinical trials were small. So, a further investigation of the association between MS symptoms and biotin should be undertaken.

### 2.4. Vitamin A

The proteins and cells involved in the immune system have specific actions and role [36]. The role of T-helper 17(Th17) cells and T regulatory (Treg) cells in MS pathogenesis, the effect of vitamin A, and of its active metabolite retinoic acid (RA), as well as the management of inflammation, have been well analyzed, mainly in in vitro studies. Also it is known that in MS, the balance between Th17 cells and Treg cells is diminished [36]. Research has shown through magnetic resonance imaging (MRI) that serum-retinol can predict new T1Gd (+) and T2 lesions 6 months ahead. Notably, the increase of retinol by 1μmol is likely to decrease the risk of developing Gadolinium (Gd) enhancing lesions, new T2 lesions and active lesions by 49%, 42% and 46%, respectively. Gd-enhancement is a marker for blood brain barrier breakdown, and histologically is correlated with the inflammatory phase of lesion development [37].

Vitamin A may amend MS pathogenesis via several mechanisms. Those mechanisms include the reduction of inflammatory processes by re-balancing pathogenic (Th1, Th17, Th9) and immune-protective (Th2, Treg) cells, modulating the B-cell and dendritic cell functions, as well as increasing autoimmunity and regeneration tolerance in the CNS [38]. Thus, vitamin A could be considered as a potential co-treatment agent in MS disease management [38].

Several clinical studies have shown that active vitamin A derivatives may suppress the formation of pathogenic T-cells in MS patients. Notably, a recent study by Bitarafan et al. determined the effect of vitamin A on disease progression [17]. 101 RRMS patients participated in a placebo-controlled randomized clinical study. The treated group received 25,000 IU per day retinyl palmitate for 6 months followed by 10,000 IU per day retinyl palmitate for another 6 months. The results of the EDSS and MS functional composite (MSFC) were recorded both at baseline and at the end of the study. Moreover, the relapse rate during the intervention period was recorded. It was shown that MSFC was significantly improved in the treatment group, whereas there were no significant differences between EDSS changes in the treated and placebo groups. Enhanced brain active lesions were recorded in both groups. Moreover, no significant difference in the volume of T2 hyper-intense lesions between the two groups after intervention was recorded. Thus, this clinical study supported evidence that vitamin A may improve total MSFC score in RRMS patients, however it cannot affect EDSS, relapse rate and brain active lesions [17]. A year later, Bitarafan et al. applied a modified scale of effects based on fatigue and Beck Depression Inventory-II scales at both at the beginning and the end of a one year study [18]. Notably, a significant improvement in the treated group for both fatigue and depression was found. In view of the above, this study supported evidence that vitamin A supplement may contribute to interferon therapy, improving also mental health outcomes [18].

Eriksen et al. explored the potential effects of RA on B-cells of 25 female RRMS patients [39]. This clinical study showed that B-cells derived from MS patients, which were co-stimulated via the toll-like receptors (TLRs), TLR9 and RP105, secreted lower levels of the anti-inflammatory cytokine IL-10 compared to B-cells derived from 15 female healthy controls. Importantly, it was found that RA increased IL-10 secretion by MS-derived B-cells without influencing the levels of the pro-inflammatory cytokine TNF-α. Moreover, RA exerted the same ability to induce IL-10, as well as interferon-β-1b (IFN-β-1b). The B-cells of MS patients treated with glatiramer acetate or IFN-β-1b still displayed the beneficial effects of RA on the IL-10/TNF-α ratio [39].

Vitamin A is a fat-soluble molecule and its long-term intake in high doses may exert some adverse effects. In this aspect, Jafarirad et al. investigated its possible complications as well as potential solutions to minimize its adverse effects [40]. In this, double blind, randomized clinical trial, vitamin A, as retinyl palmitate, was administrated to 35 RRMS patients in order to investigate whether it can regulate their immune system with a dose of 25,000 IU/day for a period of 6 months. Lipid profiles, fasting blood sugar and liver enzymes were assessed to investigate the possible biochemical adverse effects [40]. It was found that vitamin A did not affect lipid profiles, fasting blood sugar and liver enzymes in both groups [40].

Collectively, although the currently available clinical studies have supported preliminary evidence for the potential beneficial effects of vitamin A supplementation as part of MS treatment, there is at present no clear clinical indication for vitamin A supplementation as a co-treatment for MS disease management. Consequently, further clinical trials focusing on vitamin A, as a potential supplementation or as an add-on option are strongly recommended, also focusing on the distinct MS phenotypes as distinct disease subpopulations.

### 2.5. Vitamin B12

Vitamin B12 exerts a crucial role in myelin synthesis and integrity [1]. A high prevalence of low serum and/or cerebrospinal fluid vitamin B12 levels in MS patients has been observed, which may be ascribed to a disorder of vitamin B12 binding or transport [1]. MS has been sometimes associated with abnormalities of vitamin B12 absorption or utilization, however, the association of these abnormalities with the disease process remain unclear [1]. Wade et al. in a placebo controlled, randomized study, determined the effect of vitamin B12 on disease progression in MS patients. A total of 138 patients with clinically definite MS, measurable disability on Guy’s neurological disability scale (GNDS), without relapse in the preceding 6 months and without antidepressant drugs prescription were enrolled. All MS patients received 1 mg vitamin B12 intramuscularly on a weekly basis, and either 70 mg lofepramine and 500 mg L-phenylalanine twice daily, or placebo tablets for 24 weeks. At the end of the treatment, there was no statistically significant difference between the groups at baseline or at follow up. MS patients improved by 2 GNDS points after treatment with vitamin B12 injections, while the addition of lofepramine and L-phenylalanine resulted in a 0.6 points benefit [41].

However, Booth et al. did not observe any neurological improvement in 26 patients during their study in which 100 μg of vitamin B12 were intramuscularly administered every other day for 3 months. However, improvements on appetite and overall well-being in 25% of the patients were recorded. The authors therefore suggested that the positive effect to vitamin B12 may be ascribed to a placebo effect or to a carryover effect from the previous vitamin B12 treatment [1]. Kira J et al. studied the serum vitamin B12 levels and unsaturated vitamin B12 binding capacities in 24 MS patients. No decrease in vitamin B12 levels was noted, however, a considerable reduction in the unsaturated vitamin B12 binding capacity was found. Afterwards, they administered a great dose of 60 mg methyl vitamin B12 every day for 6 months to 6 patients with chronic progressive MS. The motor disability was not clinically improved, however, the abnormalities in both the visual and the brainstem auditory were improved during therapy compared to the pre-treatment period. Hence, a high dose of methyl vitamin B12 may be beneficial as complementary therapy to immunosuppressive treatment for chronic progressive MS [42].

There is an urgent need for larger studies to determine the role of vitamin B12 supplementation alone, or in combination with other therapeutic agents, in prevention or reversal of MS, and aid in improved quality of life of MS patients.

### 2.6. Lemon Verbena

Inflammation is one of the central aetiological and pathogenetic factors of MS. Dietary interventions with Lippia citriodora (lemon verbena) extracts have been found to be effective in inflammatory disease prevention [19]. In fact, Mauriz et al. evaluated the effect of lemon verbena supplementation in pro- and anti- inflammatory serum biomarkers of patients with different clinical subtypes of MS in a randomized double-blinded placebo-controlled study with 30 MS patients. Serum cytokine and C-reactive protein (CRP) levels were assessed in both groups for each clinical subtype of MS. After 28 days of supplementation CRP concentrations were considerably lower in SPMS patients compared to the placebo group. Also, IFN-γ levels decreased for all treatment groups, whereas reduced IL-12 levels for RRMS patients were noted. Anti-inflammatory cytokine concentrations of IL-4 and IL-10 increased in SPMS patients [19].

The above results are seem interesting, but need new clinical trials with more MS patients.

### 2.7. Melatonin

Melatonin is a neurotransmitter released via circadian rhythm by the brain to facilitate sleep [43] has a possible neuroprotective role in processes that can lead to the demyelination and inflammatory response typically observed in MS. The less melatonin one produces, the higher frequency of symptoms and even relapses can be observed [43].

Several studies have reported that the QOL is lower in MS patients compared to healthy individuals due to increased prevalence of sleep disturbances, fatigue and depression [44]. Adamczyk et al. evaluated the effect of 5 mg melatonin daily supplementation for 90 days on serum MDA concentration, SOD activity, as well as their influence on the QOL of MS patients. 102 MS patients and 20 healthy, age- and sex-matched controls participated in this study. MDA concentrations increased in all MS patients groups, yet after treatment MDA concentrations decreased significantly in the interferons-beta and glatiramer acetate-treated groups, but not in the mitoxantrone-treated group. An increase in SOD activity was only observed in the glatiramer acetate-treated group, when compared to controls. After 3-months melatonin supplementation, SOD activity increased compared to the initial levels in the interferon beta-treated groups. Considerable increases in the mean MSIS (Multiple Sclerosis Impact Scale) -29-PHYS (Physical) and MSIS-29-PSYCH (Psychological) item-scores in the mitoxantrone-treated group were noted in relation to the other groups. No significant difference concerning mean MSIS-29-PHYS was noted before and after treatment. Conclusively, melatonin supplementation led to a reduction in mean MSIS-29-PSYCH scores compared to initial score in the interferon beta-treated groups [44].

Golan et al. investigated the changes on vitamin D and melatonin in MS patients in a randomized, double blind study with 40 IFN-β treated MS patients. 21 patients were treated with 800 IU of vitamin D3 per day (low dose), while 19 patients received 4370IU vitamin D3 per day (high dose) for a period of 1-year [45]. After a 3-month long supplementation, vitamin D levels increased and night-time melatonin secretion was considerably reduced in the high vitamin D dose group, but not in the other group. After 12 months of supplementation, a decrease in vitamin D levels, and an increase of urine nighttime 6-SMT were noted in the high dose group. Percent change in serum vitamin D was negatively associated with percent change in urine 6-SMT after 3 months, and between 3 months to 12 months. Hence, melatonin may be considered to be a possible mediator of vitamin D neuro-immunomodulatory effects in this patient group [45].

### 2.8. L-Carnitine and Acetyl-L-Carnitine

Carnitine has a crucial role in energy production by assisting the transport of fatty acids into mitochondria [46]. ALC functions as a neurotransmitter [46]. In clinical studies, L-carnitine administration improved medication-induced fatigue in MS patients, and ALC treatment improved MS-related fatigue [46]. In a prospective open-labelled study conducted on 170 MS patients, of whom 80% suffered from fatigue, and 70 healthy controls, Lebrun et al., found that MS patients receiving immunosuppressive drugs presented considerably higher mean serum carnitine concentration compared to healthy individuals. Drug-treated patients presenting decreased serum carnitine levels received a daily dose of 3–6 g of L-carnitine. After 3 months, 63% of patients experienced an improvement in fatigue, and especially those receiving cyclophosphamide or interferon treatment [46].

In another double-blinded clinical study [47], 36 patients presenting MS-related fatigue randomly received ALC at a dose of 1g twice daily or amantadine at a dose of 100 mg twice daily for a period of 3 months. After a 3-month washout period, the alternate treatment for an additional 3-month period was administered to each patient. A trend for fatigue improvement, assessed by a lower score on the FSS, was found in 70% of patients receiving ALC treatment, and in 43% of patients receiving amantadine treatment. ALC treatment considerably improved the mean FSS score compared to amantadine treatment, possibly via a nonsignificant improvement during ALC treatment combined with a worsening of the mean score during amantadine treatment. However, it is important to note that there were dropouts due to side effects [47]. Moreover, the study by Tomassinia et al, supported evidence that ALC may be more effective and better-tolerated compared to amantadine, as far as the treatment of MS-related fatigue is concerned. However, a recent study showed that amantadine treatment for a period of 1 month improved fatigue in RRMS patients as assessed by modified fatigue impact scale (MFIS) [20]. The aim of this study was to compare the efficacy of amantadine, modafinil and ALC with placebo in MS patients. More to the point, 60 MS patients with a disability level ≤5.5 on the Kurtzke EDSS and fatigue were included in the study. Patients were assigned to a 1 month treatment with either 200 mg amantadine, 2 g ALC, 200 mg modafinil or placebo. The MFIS was applied to evaluate the treatment efficacy. A considerable decreased mean MFIS score was observed after 1 month in patients treated with amantadine compared to those treated with placebo. A non-significant lower MFIS score in ALC group in comparison to placebo was recorded and the QOL of patients proved to be considerably improved due to amantadine treatment [20].

### 2.9. Vitamin D

Vitamin D constitutes an essential fat-soluble vitamin obtained through exposure to sunlight as well as dietary sources such as animal protein, fish liver oil, and fortified dairy and cereal products [48]. Adults are recommended to consume 600 international units (IU) of vitamin D per day to maintain adequate vitamin D status, which is defined as serum 25 hydroxyvitamin D (25(OH)D) concentrations greater than 75 nmol/L [48]. Vitamin D affects gene transcription through interaction with the vitamin D receptor on cell membranes. Most of these genes are related to mineral metabolism, reflecting vitamin D’s role in bone mineral homeostasis, but it also performs other functions, including its role in cell differentiation, proliferation, and growth. Specific to the immune system, vitamin D assumes paracrine hormone functions that support the maintenance of immunity, and reduce inflammation [49]. The exact mechanism of vitamin D’s potential therapeutic role in MS remains unclear, but it is suggested that it may be tied to the purported immunologic benefits, and reduced breakdown of nervous system tissue [49]. Small clinical studies of vitamin D supplementation in MS patients have documented beneficial effects on the immune system, the relapse and the gadolinium-enhancing lesions [50]. However, further studies are needed.

Recent studies have suggested that enhanced antibody reactivity against Epstein–Barr nuclear antigen-1 (EBNA1) and viral capsid antigen (VCA) may be related with high MS risk [51]. Najafipoor et al. investigated whether the immune response against latent Epstein–Barr virus (EBV) infection could be affected by vitamin D3 supplementation in MS patients. In this clinical study, 40 RRMS patients were enrolled [51]. All RRMS patients were seropositive for EBV prior to vitamin D supplementation. In this cohort, 22.5% and 47.5% of the MS patients were either deficient in vitamin D and had insufficient vitamin D levels, respectively. In a second sampling, taken after vitamin D supplementation, 27 patients were further supplemented with vitamin D3 at a dose of 50,000 U/week for 6 months and 30 enrolled as controls. 25-hydroxyvitamin D levels and immunoglobulin G titters against EBNA1 and VCA were assessed before and after supplementation [51]. The findings of this study confirm that antibody titers against EBV in MS patients rise after the onset of the disease and indicate that vitamin D3 supplementation could limit augmentation of these titers in MS patients [51].

Seasonal fluctuations in solar radiation and vitamin D levels may regulate the immune response against EBV infection, affecting the subsequent MS risk. Lossius et al, undertook a case control study of Environmental Factors In MS (EnvIMS), where 1660 MS patients and 3050 controls participated from Norway and Italy and reported the season of past infectious mononucleosis (IM). The authors observed that IM was associated with MS independently of spring season, when vitamin D levels are lower. The distribution of IM may help investigate the association between solar radiation or other factors with a similar latitudinal and seasonal variation [52].

Kubicka et al. evaluated the concentration of vitamin D3 and indices of Calcium and Phosphorus metabolism at different times of RRMS [53]. 30 RRMS patients, 15 at the early stage and 15 at the advanced stage of MS took part and underwent neurological assessment according to the EDSS [53]. The results were compared to a 15-subject control group which was matched to the age, residence, ethnicity and sex of the patients. Considerably lower serum vitamin D3 concentrations in MS patients were observed compared to the controls, while MS patients and especially women at the advanced stage of the disease had lower vitamin D3 concentrations than patients at the early stages of MS [53].

Toghianifar Ν et al. assessed the effect of high oral dose of vitamin D on IL-17 levels in 94 RRMS patients in a double blind randomized clinical study. Treatment group received 50,000 IU vitamin D3 every 5 days for 12 weeks and the other group received a placebo. Both groups were receiving interferon-β (IFN-β) treatment [21]. Serum IL-17 levels were determined at baseline and after 12 weeks of intervention. IL-17 levels presented a considerable change in RRMS patients after a high vitamin D3 dose. The multiple linear regression analysis showed that vitamin D3 intake was positively associated with IL-17 levels, after adjustment with EDSS scores [21].

MS symptoms are related with enhanced production of inflammatory cytokines and reduced production of some anti-inflammatory cytokines. Mahon et al. observed the cytokine profile in MS patients following vitamin D supplementation [54]. More to the point, 39 MS patients were enrolled in a double-blinded study, and were randomized to either receive 1000 IU vitamin D and 800 mg Calcium (*n* = 17) or placebo with 800 mg Calcium (*n* = 22) daily for 6 months. Calcium was administered to all patients to prevent Calcium deficiency, which may cover any potential effect of vitamin D. As expected, serum 25(OH)D levels increased significantly after the 6 months supplementation. Vitamin D supplementation also significantly increased serum (TGF)-h1 levels within 6 months, whereas the placebo treatment did not exert any effect on serum TGF-h1 levels. Also, TNF-a, IFN-g, and IL-13 levels did not alter following vitamin D supplementation and the IL-2 mRNA levels were reduced, following vitamin D supplementation at a no significant level, though [54].

In addition, Burton et al. performed an open-label randomized controlled 52-week clinical study to explore the tolerability of a high-dose oral vitamin D supplement and its effect on biochemical, immunologic, and clinical outcomes in 25 MS patients and 24 healthy individuals. The treatment group was administered with increasing vitamin D dosage up to 40,000 IU daily over a 28-week period, followed by 10,000 IU daily for 12 weeks, and further down-titrated to 0 IU daily/day. Also, throughout the clinical study, 1200 mg/day Calcium were administered, even though Calcium-related levels between the two groups were normal. It was confirmed that 10,000 IU/day of vitamin D in MS was safe, with evidence for immunomodulatory effects. Serum Calcium levels were not significantly increased by high dose vitamin D supplementation [55]. However, this clinical study had several drawbacks concerning statistical analysis and design. Furthermore, both T-cell reactivity and proliferation was reduced considerably in the patient group over the 52-week period, while no change was found in healthy individuals. This reduction was more evident in patients who retained vitamin D levels at 100 nmol/L at 52 weeks [55]. The MMP-9/TIMP-1(TIMP metallopeptidase inhibitor 1, or TIMP1 values were altered over the clinical trial, however, the nature and the magnitude of change were not differentiated between groups. Moreover, cytokine profiles did not result in significant patterns of change, possibly due to the lack of influence of vitamin D on those markers, the impact of confounding factors, as well as the existence of methodological limitations [55].

Aivo et al. evaluated the impact of weekly vitamin D3 supplementation on serum levels of various cytokines in RRMS patients [56]. All patients were treated with IFN-beta-1b and were randomized to a supplementary treatment with either 20,000 IU/week cholecalciferol or placebo. Concentrations of LAP (TGF-β), INF-γ, IL-17A, IL-2, IL-10, IL-9, IL-22, IL-6, IL-13, IL-4, IL-5, IL-1β and TNF-α were assessed at baseline and at 12 months. TGF-β levels were increased considerably in the vitamin D treated group at 12 months. Also, placebo treatment did not exert considerable impact on LAP levels. The levels of the other cytokines were not considerably altered in any group. Hence, the immune regulatory effects of TGF-β could contribute to the improvement of MRI outcomes in the patients treated with vitamin D [56].

Muris et al. performed a clinical study in which 30 RRMS patients received IFNβ-1a with high vitamin D3 dose, and 23 patients received placebo for 48 weeks [57]. Lymphocytes were phenotypically determined by flow cytometry and in vitro cytokine secretion was evaluated in the presence or absence of vitamin D3 by the use of Luminex technology. High vitamin D3 dose did not lead to any increase in lymphocytes with a regulatory phenotype. However, this study supported the hypothesis that vitamin D may contribute to immune homeostasis maintenance by suppressing the disruption of the T-cell compartment early in the disease course of MS [57]. A year later, Muris et al. investigated whether Vitamin D status could influence the disability progression of MS patients in a retrospective 3-year study conducted on 554 MS patients [58]. Baseline vitamin D status was not significantly correlated with either disability or disability progression, regardless of MS phenotype. This study supported evidence that within the normal range, 25(OH)D status may be related with relapse occurrence in younger MS patients, however, it may not be correlated with disability or disability progression over a 3-year follow-up [58].

Patients with MS are in high risk of osteoporosis and fractures [59]. Poor vitamin D status is considered to be a risk factor for MS, and vitamin D supplementation has been recommended to both inhibit MS progression and retain bone health [59]. Steffensen et al. in their double-blinded study assessed whether a weekly dose of 20,000 IU vitamin D3 may suppress bone loss in ambulatory MS patients [60]. In this study, 71 MS patients participated and were randomised to receive 20,000 IU vitamin D3 or placebo once for a week and 500 mg Calcium daily for 96 weeks. The mean serum 25(OH)D levels in the intervention group increased from 55 nmol/L baseline to 123 nmol/L at 96 weeks, and after the study period percentage change in Bone Mineral Density did not differ between groups at any site. Furthermore, the BMD was reduced at the hip in the treatment group [60].

Flu-like symptoms constitute a common side effect of IFN-β) treatment in MS patients and are associated with post-injection cytokine surge. Golan et al., assumed that vitamin D3 supplementation may ameliorate the symptoms by reducing related serum cytokines’ levels [61]. In this aspect, 45 IFNβ-treated patients were randomized to receive either 800 IU/ day (*n* = 21 patients) or 4370 IU/day (*n* = 24 patients) of vitamin D for 1 year. The serum levels of 25-OH-D, Calcium, Parathyroid hormone, IL-17, IL-10, and IFN-γ were evaluated periodically and EDSS, relapses, adverse events and QOL were assessed. Vitamin D supplementation in IFN−β treated patients was safe, and it was associated with dose dependent changes in IL-17 serum levels, but not with IFN−β related flu-like symptoms [61].

Achiron A. et al., evaluated the impact of alfacalcidol, a vitamin D analogue, on MS-related fatigue, in a randomized, double-blinded, placebo-controlled clinical study in MS patients. 600 randomly selected patients from the Sheba MS Registry database were evaluated using the FSS. 259 patients with clinical fatigue were additionally evaluated for trial eligibility, and 158 MS patients with significant fatigue were included in the study and were randomized to receive either 1 mcg/day Alfacalcidol (*n* = 80 patients) or placebo (*n* = 78 patients) for 6 months [22]. Alfacalcidol resulted in a decreased mean relative Fatigue Impact Scale score as compared to placebo. This positive effect was further highlighted when relative change was estimated. QOL improved in Alfacalcidol-treated patients compared to the patients that received the placebo in the RAYS psychological and social sub-scales. Notably, the Alfacalcidol-treated group presented reduced number of relapses and a higher proportion of relapse-free patients [22]. Furthermore, the Alfacalcidol-related decreased relapses reached statistical significance at 4 months of treatment and was continued at 6 months but stopped 2 months after discontinuation. Treatment with Alfacalcidol was safe, while no serious adverse effects were noted [22]. Thus, Alfacalcidol could be a safe and effective treatment strategy to reduce fatigue and increase QOL in this study population.

Nevertheless, the impact on clinical aspects is inconclusive, especially when the influence of supplementation is assessed. The effect of vitamin D on immune cells subsets in relation to clinical studies should be studied more.

### 2.10. Lipoic Acid

Enhanced levels of oxidative stress are implicated is several inflammatory processes, exerting crucial role in MS pathogenesis [62]. In their double-blind, randomized controlled clinical study, Khalili et al. examined the impact of daily consumption of lipoic acid (LA) on oxidative stress among MS patients [62]. More to the point, 52 RRMS patients, aged 18–50 years with EDSS ≤ 5.5 were assigned to consume either 1200 mg/day LA or placebo capsules for 12 weeks. LA intake led to a considerable improvement of TAC compared to the placebo group. A considerable change of TAC was found within the treatment group, however, other markers of oxidative stress were not affected by LA [62].

Yadav et al. examined the pharmacokinetics, tolerability and effects on MMP-9 and Soluble intercellular adhesion molecule-1 (sICAM-1) after oral administration of LA in MS patients [63]. Thirty-seven MS patients were randomly assigned to 1 of 4 groups: Placebo, 600 mg LA twice a day, 1200 mg LA once a day or 1200 mg LA twice a day for 14 days. Patients taking 1200 mg LA had considerably higher peak serum LA levels compared to those receiving 600 mg and those peak levels varied among subjects. A considerable negative association between peak serum LA levels and mean changes in serum MMP-9 levels was noted. There was also a considerable dose-response relationship between LA and mean change in serum SICAM-l levels, while LA was generally well-tolerated and may reduce serum MMP-9 and sICAM-1 levels. Thus, LA may consider useful in treating MS by suppressing MMP-9 activity and interfering with T-cell migration into the CNS [63].

The intake of Lipoic Acid as a dietary supplement in MS patients is less researched and documented.

### 2.11. Folic Acid

Isager H. examined the impact of folic acid in MS. In particular, 21 MS patients participated, of whom 5 had subnormal serum folate levels. It remains unclear whether folate deficiency can worsen MS symptomatology. Folate deficiency has a negative effect on overall health and the neurological system, folate status should be assessed in MS patients and in case of deficiency supplementation is essential [64].

## 3. Conclusions

Nutritional status of MS patients, as well as dietary supplementation have been suggested as potential crucial factors affecting both disease risk and progression. Several substantial studies have documented a systematically high prevalence of vitamin A, B12 and D3 deficiency amongst MS patients. At present, clinical data suggest that most of the dietary supplements under study may exert antioxidant and anti-inflammatory properties, improving depression symptomatology and QOL, overall. However, malnutrition risk in MS patients has not been adequately explored in order for more precise conclusions to be drawn. On the other hand, physical activity seems to be a critical factor in the amelioration of MS symptomatology and improvement of psychological status and life expectancy.

Overall, several dietary supplements may decrease inflammation and fatigue, while increasing autoimmunity tolerance in MS patients, and thus improving QOL and life expectancy. However, it should be emphasized that there is no effective clinical indication, so far, for applying dietary supplementation as complementary treatment against MS symptomatology. Further clinical trials focusing on each dietary supplementation separately as potential complementary therapeutic agent should be performed in this sensitive patient subpopulation.

However, future prospects and objectives in research about the clarification of the mechanisms and molecular pathways that promote or inhibit the pathophysiology of the disease should take into account the following parameters:The evaluation of the composition of the gut microbial load;The assessment of the defects of the intestinal immune system;The clarification of the role of metabolism of polyphenols and Vitamin D;The study the effect of dietary agents, extracts and drugs on the signaling pathways involved in AMPK/SIRT/PPAR cascade or to study the NF-kB transcription factor in the light of the disease;The identification of possible interactions between complementary dietary interventions and medication agents taken by the patient with MS.;The establishment of a committee to organize a campaign to inform and educate patients about the importance of maintaining a healthy dietary pattern during treatment.

## Figures and Tables

**Figure 1 medicines-06-00095-f001:**
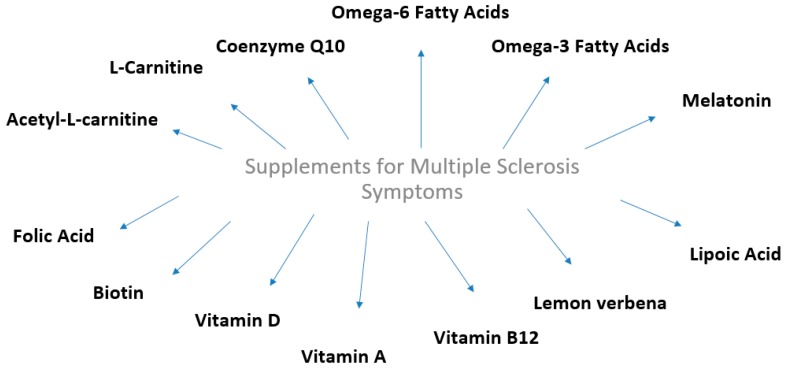
Dietary supplements that exert beneficial effects against multiple sclerosis symptoms and relapses.

**Table 1 medicines-06-00095-t001:** Supplements that present beneficial effects against multiple sclerosis symptoms and relapses.

Dietary Supplement	Study Type	Measured Parameters	Number of Patients and Type of MS	Supplement Administration	Effects	References
FA	Double-blind control clinical study	Relapse rates	116 MS	2 groups received linoleic acid, and 2 control groups received oleic acid	20 g linoleic acid marginally affected the duration and severity of relapses of MS but had no effect on overall disability	[8]
Ω-3 FA	Double-blind control clinical study	Kurtzke Disability Status Scale Score	112 RRMS	10 g/day of FO and diet or placebo OO and diet	After 2 years, 51% of patients in the FO group and 41.4% of those OO group showed improved or unchanged scores, according to the Kurtzke Disability Status Scale	[9]
Ω-3 FA	Double-blind clinical study	Qol questionnaire: Neurological status and relapse rate	31 RRMS	1 group received a low fat diet (15% fat) with FO and 1 group received the AHA Step I diet (fat 30%) with OO	Decreased fatigue on the OO group at 6 months. Both groups had reduced relapse rates compared to the rates during the 1 year prior to the study	[10]
Ω-3 FA	Open-label designed clinical study	Immune cell secretion of MMP-9	10 RRMS	ω-3 FA for 3 months	ω-3 FA decreased MMP-9 levels, while their immune cell secretion of MMP-9 was considerably reduced by 58% after 3-months and a significant increase in ω-3 FA levels in red blood cell membranes was recorded	[11]
CoQ10	Controlled randomized double-blinded clinical study	Inflammatory markers (TNF-α, IL-6, and MMP-9)	48 RRMS	500 mg CoQ10/day and or placebo for 12 weeks	CoQ10 supplementation at a dose of 500 mg/day may improve fatigue and depression in MS patients	[12]
CoQ10	Controlled randomized double-blinded clinical study	TNF-α levels	48 RRMS	500 mg CoQ10/day and or placebo for 12 weeks	TNF-α levels decreased significantly in the CoQ10 group. CoQ10 supplementation also resulted in decreased serum levels of MMP-9 as compared to the placebo group	[13]
CoQ10	Controlled randomized double-blinded clinical study	MDA, TAC and antioxidant markers (SOD, GPx)	48 RRMS	500 mg CoQ10/day and or placebo for 12 weeks	Decrease of oxidative stress and increase antioxidant enzyme activity in RRMS	[14]
Vitamin B7 (Biotin)	Uncontrolled, non-blinded proof of concept study	Quantitative and qualitative measures: Visual actuality, magnetic resonance spectroscopy (H-MRS) of the Choline/Creatine ratio, disability and progression in progressive MS.	23 MS	100–300 mg/day biotin for a period from 2 to 36 months	High biotin doses exerted a positive effect on disability and progression in this MS patient population	[15]
Vitamin B7 (Biotin)	Double-blind, placebo-controlled study	(EDSS) score: Reversal of MS-related disability.	154 PRMS	MD1003 (biotin 100 mg) or placebo orally thrice daily	Reduction EDSS progression and improved clinical impression of change compared with placebo	[16]
Vitamin A	Controlled randomized clinical study	Relapse rate: (EDSS) and (MSFC)	101 RRMS	25,000 IU/dretinyl palmitate for 6 months followed by 10,000 IU/d retinylpalitate for another 6 months	Reduction of progression of disability, upper limb and cognitive functions	[17]
Vitamin A	Controlled randomized clinical study	Modified fatigue impact scale and Beck Depression Inventory-II (fatigue and depression)	101 RRMS	25,000 IU/dretinyl palmitate for 6 months followed by 10,000 IU/d retinyl palitate for another 6 months	Vitamin A improved the depression through the modulation of inflammatory conditions	[18]
Lemon verbena	Randomized double-blinded placebo-controlled study	Serum levels of C reactive protein and 8 cytokines/ inflammatory markers (IFN-γ, IL-12, IL-23, IL-6, TNF-α, TGF-β, IL-4 and IL-10)	30 MS	Lemon verbena supplementation (10% w/w verbascoside)	After 28 days, CRP concentrations were considerably lower in SPMS patients compared to the placebo group, IFN-γ levels decreased for all MS-treated groups, whereas reduced IL-12 levels for RRMS patients were noted. Anti-inflammatory cytokine concentrations of IL-4 and IL-10 increased in SPMS patients	[19]
ALC	Pilot randomized, blind clinical study	Kurtzke Expanded Disability Status Scale (EDSS) and fatigue	60 MS	A 1 month treatment with either 200 mg amantadine, 2 g ALC, 200 mg modafinil or placebo	The amantadine treatment for a period of 1 month improved fatigue in RRMS patients as assessed by MFIS	[20]
Vitamin D	Controlled randomized double-blinded clinical study	IL-17 levels	94 RRMS	Received 50,000 IU vitamin D3/5 days for 12 weeks or placebo	IL-17 levels showed significant change in RRMS patients after receiving high dose vitamin D3 for 12 weeks	[21]
Alfacalcidol	Controlled randomized double-blinded clinical study	Fatigue Impact Scale (FIS) score	600 MS	80 patients received alfacalcidol (1 mcg/) and 78 patients placebo for 6 months	QoL improved in Alfacalcidol-treated patients as compared with placebo. The Alfacalcidol-treated group had reduced number of relapses and higher proportion of relapse-free patients	[22]

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
