# Peer review of "Dietary Supplements on Controlling Multiple Sclerosis Symptoms and Relapses: Current Clinical Evidence and Future Perspectives"

_medicines, 2019, doi:10.3390/medicines6030095_

Round 1
Reviewer 1 Report
This is a valuable review of dietary supplementation in MS. The section on melatonin stood out as being a medication rather than a dietary supplement. The other agents that were included are of interest because these are used by patients. I wonder if the authors might consider adding curcumin which is of great interest to some patients
The text of the paper gives useful background for the use of the supplements and the biochemical processes that are of interest.
The authors present a useful table that summarized the clinical trials.
It would be of interest to have a meta-analysis of the results for the different supplements
My main comment is that the authors could be more critical of the clinical trials. The reader needs to know whether the authors think the results are sound and whether the trials were well-conducted. Compared to trials of MS therapies, these are small trials and it is suprising to see positive results with small numbers. This could be discussed more, and the authors could comment on the quality of the studies.
In the discussion the authors talk about dietary supplements reducing inflammation. Do they mean that the papers that have been reviewed demonstrate this? Or is this a more general statement in which case it needs to be backed up with references from the literature.
Given the current interest in the gut microbiota in MS, it would be useful to have some comments about this topic, and whether the dietary supplements might influence the gut microbiota.
Author Response
Dear Sir/Madam,
Thank you very much for taking into account and for reviewing our manuscript.
Below, I attach our responses to the reviewers’ comments.
Kind regards,
Christina Tryfonos
We tried to correct the English in the review and also all the grammatical mistakes. Also, we tried to correct and reduce plagiarism.
Response to Reviewer 1
-My main comment is that the authors could be more critical of the clinical trials. The reader needs to know whether the authors think the results are sound and whether the trials were well-conducted.
We have added paragraphs and we discuss more the quality of the studies in the end of each subchapter. We have added the following section:
Further research on these supplements and the MS should be continued with new and bigger clinical trials. The above results are seem interesting, but they need more clinical trials on a larger population on MS patients.
A lot of benefits have been observed after biotin supplementation in MS patients but the clinical trials were small .So a further investigation of the association between MS symptoms and biotin should be undertaken.
Collectively, although the currently available clinical studies have supported preliminary evidence for the potential beneficial effects of vitamin A supplementation as part of MS treatment, there is at present no clear clinical indication for vitamin A supplementation as a co-treatment for MS disease management. Consequently, further clinical trials focusing on vitamin A, as a potential supplementation or as an add-on option are strongly recommended, also focusing on the distinct MS phenotypes as distinct disease subpopulations.
There is an urgent need for larger studies to determine the role of vitamin B12 supplementation alone, or in combination with other therapeutic agents, in prevention or reversal of MS, and aid in improved quality of life of MS patients.
The above results are seem interesting, but need new clinical trials with more MS patients.
Nevertheless, the impact on clinical aspects is inconclusive, especially when the influence of supplementation is assessed. Τhe effect of vitamin D on immune cells subsets in relation to clinical studies should be studied more.
The intake of Lipoic acid as a dietary supplement in MS patients is less researched and documented.
Reviewer 2 Report
The paper entitled "Dietary supplements on controlling multiple sclerosis symptoms and relapses: Current clinical evidence and future perspectives" is a very interesting review, it certainly can be published in Medicines after correction and clarification of the following details.
The abstract is very long and some parts should be rewritten (specifically the results parts). In addition, the conclusions section is a copy of part of the abstract. These details should not occur in the manuscripts. So, both section must be rewritten. The introduction section is a bit vague. Many details about the disease are mentioned, which are interesting. However, the importance of diet or dietary supplementation in relation to disease prevention and development is not just mentioned in the introduction. Given the theme of the review, these aspects must be introduced to properly focus the objective of the manuscript. The organization of the sections describing the studies is well done and clear. The Table helps a lot in the follow-up of the review. As a suggestion, the parameters measured in each study should be included in the table. In addition, many data are repeated in the table and in the text. Check this. I miss an exhaustive discussion about future perspectives related to the aim of the review. I suggest to include a new section to discuss about it.Author Response
Dear Sir/Madam,
Thank you very much for taking into account and for reviewing our manuscript.
Below, I attach our responses to the reviewers’ comments.
Kind regards,
Christina Tryfonos
We tried to correct the English in the review and also all the grammatical mistakes. Also, we tried to correct and reduce plagiarism.
Response to the Reviewer 2
-The abstract is very long and some parts should be rewritten (specifically the results parts). In addition, the conclusions section is a copy of part of the abstract. These details should not occur in the manuscripts. So, both section must be rewritten.
We have corrected the abstract:
Multiple sclerosis (MS) constitutes a chronic progressive demyelinating disease which negatively affects the central nervous system. MS symptoms detrimentally affect the quality of life, as well as the life expectancy of MS patients. In this aspect, the present study aims to critically summarize and evaluate the currently available clinical studies focusing on the potential beneficial effects of dietary supplements on controlling MS symptomatology and relapse. Methods: PubMed database was comprehensively searched, using relative keywords to identify clinical trials that investigated the beneficial effects of dietary supplementation against MS symptomatology and progression. 40 clinical trials were found, which were divided into categories. Results: Nutritional status of MS patients, as well as supplementation have been suggested as potential factors affecting progression. Several substantial studies have documented a systematically high prevalence of vitamin A, B12 and D3 deficiency amongst MS patients. At present, clinical data have suggested that most of the dietary supplements under study may exert antioxidant and anti-inflammatory properties, improving depression symptomatology and quality of life overall. However, malnutrition risk in MS patients has not been adequately explored in order for more precise conclusions to be drawn. The supplements that may have a positive effect on MS are vitamins, fatty acids, antioxidants, phytochemicals and melatonin. Conclusions: Several dietary supplements may decrease inflammation and fatigue, also increasing also autoimmunity tolerance in MS patients, and thus improving quality of life and life expectancy. Currently, there is no effective clinical indication for applying dietary supplementation as complementary treatment against MS symptomatology.
-The introduction section is a bit vague
We have corrected the introduction and we have added the following section:
The last years, becoming more interest the relationship between the dietary supplementation and multiple sclerosis (MS) This has focused largely on aetiology and the influence of dietary supplements as a crusial factor in the amelioration of MS symptomatology and improvement of psychological status and life expectancy. While the cause of MS remains unknown and the influence of dietary supplements is unclear, recent studies on antioxidant and dietary supplements intake in MS are strengthening the rationale in support of a healthy eating regime following diagnosis.
-I miss an exhaustive discussion about future perspectives related to the aim of the review. I suggest to include a new section to discuss about it.
We have added the following section in the conclusion:
However, future prospects and objectives in research about the clarification of the mechanisms and molecular pathways that promote or inhibit the pathophysiology of the disease should take into account the following parameters:
• The evaluation of the composition of the gut microbial load.
• The assessment of the defects of the intestinal immune system.
• The clarification of the role of metabolism of polyphenols and Vitamin D.
• The study the effect of dietary agents, extracts and drugs on the signaling pathways involved in AMPK / SIRT / PPAR cascade or to study the NF-kB transcription factor in the light of the disease.
• The identification of possible interactions between complementary dietary interventions and medication agents taken by the patient with MS.
• The establishment of a committee to organize a campaign to inform and educate patients about the importance of maintaining a healthy dietary pattern during treatment.
Round 2
Reviewer 2 Report
After the revision of the new version of the article "Clinical and Metabolic Markers Predicting Efficacy of Lisinopril Therapy”, most of my previous suggestions (modify the abstract, introduction and conclusions sections, include a section of future perspectives) have been included.
However, my suggestion about including the parameters measured by the different studies has not been made. I consider that this information would be very useful for the audience to understand the effects detected in each study carried out. Therefore, the focus of the nutritional intervention studies should be clear. For example, if they are based on parameters related to oxidative stress (Specify measured parameters), symptoms of the disease (e.g., fatigue etc,)… I recommend including this type of information in a new column of the table or in the “kind of clinical study” column.
After considering this suggestion, it can be published in Medicines
Author Response
Dear Sir/Madam,
Thank you very much for taking into account and for reviewing our manuscript.
Below, I attach our responses to the reviewers’ comments.

This manuscript is a resubmission of an earlier submission. The following is a list of the peer review reports and author responses from that submission.
Round 1
Reviewer 1 Report
This is a review article aiming to summarize the relevance of nutrition and dietary supplementation for multiple sclerosis disease progression. Although the topic is of interest, the manuscript lacks a clear hypothesis and a straight-forward discussion of the current knowledge. The introduction is very superficial and contains various wrong statements. For example, CSF examination is listed as an “alternative criteria” for MS diagnosis but indeed is essential for the diagnosis of MS. CSF analysis even help to evaluate the prognosis in patients. As another example, it is stated that Progressive MS is characterized by consistently worsening disability, however this is not the definition of progressive MS. Indeed, PPMS can be characterized at different points in time as either active (with an occasional relapse and/or evidence of new MRI activity) or not active, as well as with progression (evidence of disease worsening on an objective measure of change over time, with or without relapse or new MRI activity) or without progression. Finally, the statement “MS is more common in young adults, women, smokers, individuals who have had Epstein Barr virus” is not true but who carry EBV.
The manuscript appears to be written in a hurry. There are many typos as well as grammatical and semantical mistakes. As an example, the authors state that “It is the most common demyelinating condition of the CNS affecting an estimated 2.5 million people worldwide, 4-6 and 47 an estimated 100 thousand in Canada (3).” This sentence is wrong and I could list numerous others.
Semantic errors are for example the statements “MS diagnosis is commonly based on the McDonald criteria, which aim to determine the presence of demyelinating lesions”. None of the MS criteria require the presence of demyelination, as this is a post-mortem pathological criteria OR “It has been reported that MS patients showed significantly lower mean concentration of serum linoleic acid, lymphocytes, and cerebrospinal fluid compared to healthy individuals”. Do MS patients have less amounts of CSF? This is what the authors state!
In its current form, the manuscript is not acceptable for publication in the journal. What I suggest is to briefly state the biological and biochemical relevance of the specific compounds and then briefly summarize the main outcomes of the clinical studies. The main results could well be listed in tabular form.
Finally, a professional English proof reading service should check the manuscript before re-submission.
Reviewer 2 Report
Thank you for this interesting review on a challenging topic.
1) My main comment is that it would be better structured and easier to read if the studies were grouped and commented on by the type of outcome they used. i.e. cytokine profiles, MRI measures, MS disease outcomes (relapses or disability progression) or symptomatic outcomes (fatigue, depression, quality of life). This is because effects on cytokine profiles may be interesting and point towards potential disease mechanisms, but significant effects on fatigue (such as the alfacalcidol study) might have immediate clinical implications.
2) Furthermore, it would be useful to comment on aspects such as study size or duration, to provide an overall idea of quality. Many of the studies cited here are very small and with relatively short follow-up, but there are one or two, especially in the vitamin D section, with larger cohorts or longer follow-up, and these studies are a lot more robust. As an overall review of the literature a comment on areas where there are higher quality studies available would be of benefit.
3) With the above comment in mind, I would suggest discussing vitamin D first, as this is the area with the best quality studies and most number of studies.
4) I would also suggest having Figure 1 occur earlier in the text as it is useful to set the scene of what supplements will be discussed.
5) please check that all acronyms have been defined on first use. I could not see a definition of BMD.
6) line 407-409 “Hence, the immune regulatory effects of TGF-beta could contribute to the improvement of MRI outcomes in the patients treated with vitamin D”. It is not clear how you get from cytokine levels to MRI outcomes. Please re-word.
7) malnutrition does not seem to be discussed in the main body of the paper but is mentioned in the conclusions. Is this because there is no literature on it? if it is a relevant part of the paper, there should be a section on it in the main part of the paper.
Minor comments:
- "in fact" and "more to the point" are used in an idiosyncratic way throughout the paper. They are usually used as points of emphasis, and are a little distracting in this review. I would suggest deleting all instances of them. Otherwise I think the English is very good.
- line 412-413: I'm not sure what "High vitamin D3 dose did not lo any increase in lymphocytes with a regulatory phenotype." is supposed to mean. Is there a word missing?
Reviewer 3 Report
This review, entitled “Dietary supplements on controlling multiple sclerosis symptoms and relapses: Current clinical evidence and future perspectives”, summarizes some studies focusing on the effects of dietary supplements on multiple sclerosis symptomatology.
The structure of the review was disorganized and the message that the authors want to transmit was not clear.
Principally, the authors only described the results obtained by other authors, without giving a clear message at the end of each paragraph, summarizing and providing a conclusion on the data reviewed in each paragraph.
Some more bibliography references and a little introduction for any paragraph are necessary to introduce the theme. For example, the introduction in 2.9 paragraph was well written (lines 318-334).
A good revision of abbreviations is necessary. Remember that abbreviations only must be defined at their first mention and, please, ensure consistency of these throughout the article.
Comments:
Line 42. In line 110, “quality of life” was defined as “QOL”. Please define “quality of life” in line 42, that this is the first mention of it.
Line 87. Delete “)”.
Lines 98, 98. Missing reference.
Line 104. Define the composition of fish oil. EPA+DHA?
Lines 112, 113. Why did you use the abbreviation for fish oil and for olive oil, when earlier in the manuscript none was used? Revise lines 104 and 105.
Line 120. Explain why MMP-9 is important.
Line 121. “Quality of life” was already defined.
Paragraph 2.1.2 Omega-3 Fatty Acids. What is the conclusion of all this? Or what do you mean by this?
Line 132. “received 500 mg/day”. What did each group receive?
Line 134 “decrease of FSS”. Can FSS decrease? Or was the score of FSS to decreasing?
Line 144. “MMP-9” was already defined in line 120.
Line 145. Define TGF-β.
Line 146. “considerably”. It was significant or not?
Line 153. Delete “,” before of “dismutase”.
Lines 163, 164. Was this effect positive or negative?
Line 166. Define Th17 and write a little introduction about immune cells.
Line 167. Define RA.
Line 168. “In vitro” must be written in italic.
Line 168, 169. Explain the Th17/Treg balance relevance in MS pathology.
Line 193. Define “BDI-II scales”.
Line 218. Missing reference.
Line 249. The correct name is “Lippia citriodora”, and write it in italic.
Paragraph 2.7. A brief introduction is necessary. What is melatonin? Why MS patients use melatonin?
Paragraph 2.7 Add this case report: https://doi.org/10.1111/jpi.12203
Line 265. “SOD” was already defined in line 153.
Line 272. Define “MSIS-29-PHYS” and “MSIS-29-PSYCH item-scores”.
Line 277. Was it IFN-β?
Line 278. Why did use decimal separator? Please, be consistent along with all the manuscript.
Lines 287, 288. Missing reference.
Line 301. “FSS” was already defined in line 133.
Line 308. Define “MFIS”. You defined “MFIS” in line 312.
Line 309. “modafiniland ALCAR”. It was “modafinil and ALCAR”? “ALCAR” means “Acetyl-L-carnitine”? In line 288 you defined “Acetyl-L-carnitine” as “ALC”. Clarify this abbreviation.
Lines 318, 319. Missing reference.
Line 323. You defined “VDR” abbreviation. Why? If you do not use this abbreviation in the text, do not define it.
Lines 326, 327. Missing reference.
Line 337. Define “EBV”.
Line 343. In line 321 you defined “25- hydroxyvitamin D” as “25(OH)D”. So, why did you not use this abbreviation?
Define “IgG”.
Lines 345 – 351. I think that this paragraph was not necessary. I advise removing it.
Lines 352, 353. Why did you write “Calcium and Phosphorus” in uppercase? Revise it in the text.
Line 353. A full stop was missing after the reference (44).
Line 359. “considerably”. It was significant or not?
Lines 366, 367. IL-17 levels increased or decreased?
Line 371. “Interleukin” was already defined in line 144.
Line 377. Use “25(OH)D” abbreviation.
Lines 380, 381. Use “α” and “γ” symbols. There was not a significant change, and so? It is necessary to argue.
Line 396. Define “TIMP-1”.
Line 407. Use “TGF-β”. Define “MRI”.
Line 410. Put the full stop after the reference (49).
Line 411. “In vitro” in italic.
Line 412. Define “1,25 (OH)2 D3”.
Line 413. “not lo any”?
Line 415. Put the full stop after the reference (49).
Line 424. Missing reference.
Line 430 and 438. Use “25(OH)D” abbreviation!
Line 431. Define “BMD”.
Line 434. “IFN-β” was already defined.
Line 438. Define “PTH”.
Line 439. “Quality of life” was already defined as “QOL” in line 110.
Line 446. “Fatigue Severity Scale (FSS)” was already defined in line 133.
Line 448. Then, why did you use “mcg/mL”?
Line 450. “MFIS” was already defined.
Line 461. Missing reference.
Line 463, 464. Use “EDSS” abbreviation.
Line 466 and 467. Use “TAC” abbreviation, it was already defined in line 152.
Line 468. “lipoic acid” was already defined in line 462.
Line 469. Define “sICAM-1”.
Paragraph 2.11. A brief introduction is necessary. For example, what is folic acid?
Was this paragraph really necessary? What was the conclusion of all this? Or what did you mean by this? Revise this paragraph.
Fig. 1. It was not “melatonine”, but “melatonin”.
The “beneficial” of “Folic acid” in MS, according to what you have commented, was not clear. So, why did you integrate “folic acid” in this figure named “Dietary supplements that exert beneficial effects against multiple sclerosis symptoms and relapses”?
Lines 490, 492, 494, 497, 499. Missing references.
Lines 495-497. The review was focused on the possible beneficial effects of dietary supplements. So, why, in the conclusion, did you speak about the physical activity as a critical factor in ameliorating MS symptomatology?
Lack: “author contribution” and the declaration of “Conflicts of Interest”.
I advise a new structure for this review:
1. Introduction
2. Dietary supplements and multiple sclerosis
2.1 Fatty Acids
2.1.1 Omega-3 Fatty Acids
2.1.2 Omega-6 Fatty Acids
2.1.3 Lipoic Acid
2.2 Vitamins
2.2.1 Vitamin A
2.2.2 Vitamin B7
2.2.3 Vitamin B9 (Folic Acid)
2.2.4 Vitamin B12
2.2.5 Vitamin D
2.3 L-Carnitine and Acetyl-L-carnitine
2.4 Melatonin
2.5 Coenzyme Q10
2.6 Lemon verbena
3. Conclusion
Round 2
Reviewer 1 Report
Please carefully check the entire manuscript again for typos and semantical Errors.
Reviewer 2 Report
Thank you for your responses. I am happy with the changes you have made to my previous comments. However, my main concern is that my first three comments have not been addressed at all (copied below for reference).
1) My main comment is that it would be better structured and easier to read if the studies were grouped and commented on by the type of outcome they used. i.e. cytokine profiles, MRI measures, MS disease outcomes (relapses or disability progression) or symptomatic outcomes (fatigue, depression, quality of life). This is because effects on cytokine profiles may be interesting and point towards potential disease mechanisms, but significant effects on fatigue (such as the alfacalcidol study) might have immediate clinical implications.
2) Furthermore, it would be useful to comment on aspects such as study size or duration, to provide an overall idea of quality. Many of the studies cited here are very small and with relatively short follow-up, but there are one or two, especially in the vitamin D section, with larger cohorts or longer follow-up, and these studies are a lot more robust. As an overall review of the literature a comment on areas where there are higher quality studies available would be of benefit.
3) With the above comment in mind, I would suggest discussing vitamin D first, as this is the area with the best quality studies and most number of studies.
Reviewer 3 Report
The authors of the review titled “Dietary supplements on controlling multiple sclerosis symptoms and relapses: Current clinical evidence and future perspectives” sent a new version of this. Although they responded to different comments, the authors once again only described the results obtained by other authors without providing a clear message. A review should not be a simple recompilation of data obtained from literature, but the construction of a hypothesis and a discussion of this considering the current data.